



# Counting on Chemistry: Laboratory Evaluation of Seed Material-Dependent Detection Efficiencies of Ultrafine Condensation Particle Counters

Peter Josef Wlasits[1], Dominik Stolzenburg[1,2], Christian Tauber[1], Sophia Brilke[1], Sebastian Harald Schmitt[3], Paul Martin Winkler[1], and Daniela Wimmer[1]

[1]University of Vienna, Aerosol Physics and Environmental Physics, Boltzmanngasse 5, 1090 Vienna, Austria
[2]University of Helsinki, Institute for Atmospheric and Earth System Research/Physics, 00014 Helsinki, Finland
[3]TSI GmbH, Neukoellner Straße 4, 52068 Aachen, Germany

**Correspondence:** Paul Martin Winkler (paul.winkler@univie.ac.at)

**Abstract.** Condensation Particle Counters (CPCs) are crucial instruments for detecting sub-10 nm aerosol particles. Understanding the detection performance of a CPC requires thorough characterization under well-controlled laboratory conditions. Besides the size of the seed particles, chemical interactions between the working fluid and the seed particles also influence the activation efficiencies. However, common seed particle materials used for CPC characterizations are not chosen in respect of

chemical interactions with vapor molecules of the working fluid by default. Here, we present experiments on the influence of the seed particle material on the detection efficiencies and the 50 % cut-off diameters of commonly used CPCs for the detection of sub-10 nm particles. A remarkable set consisting of six different and commercially available particle detectors, including the newly-developed TSI V-WCPC 3789 and a tuned TSI 3776, was tested. The corresponding working fluids of the instruments are n-butanol, diethylene glycol and water. Among other materials we were able to measure detection efficiencies with

nanometer-sized organic seed particles reproducibly generated by oxidation of $\beta$-caryophyllene vapor in a flow tube. Theoretical simulations of supersaturation profiles in the condensers were successfully related to measured detection efficiencies. Our results demonstrate the importance of chemical similarities between seed particles and the used working fluids when CPCs are characterized. We anticipate our study to contribute to a deeper understanding of chemical interactions during heterogeneous nucleation processes.

## 1 Introduction

Ultrafine aerosol particles (< 100 nm) might cause severe effects on human health (Pedata et al., 2015) and impact the global climate by the aerosol indirect effect (Bauer and Menon, 2012; Albrecht, 1989). In the atmosphere, aerosol particles have primary and secondary sources. Secondary particles form when gaseous precursors oxidize to low-volatility compounds which at high enough abundances can form molecular clusters. Clusters grow by condensation into the nucleation and Aitken mode

size range and eventually reach sizes where they can act as cloud condensation nuclei (CCN). A detailed understanding of the mechanims leading to aerosol formation and growth, i.e. new particle formation (NPF, Nieminen et al. (2018)), requires the





careful measurement of particles in the cluster-particle transition and nucleation mode size regime. Quantitative measurements of aerosol particles in the size range between 1-10 nm remain a major challenge for the understanding of the mechanisms responsible for NPF.

Condensation particle counters (CPCs), which optically detect nanometer-sized particles after condensational growth, are

state-of-the art instruments for particle measurements in the sub-10 nm size regime. Since the introduction of Aitken's "Dust Counter" towards the end of the 19th century (Aitken, 1888), particle counters capable of detecting smaller and smaller aerosol particles have been developed (Stolzenburg and McMurry, 1991; McMurry, 2000; Sgro and Fernández de la Mora, 2004; Vanhanen et al., 2011; Kangasluoma and Attoui, 2019).Thus, CPCs are used in many different fields, including atmospheric studies (Brilke et al., 2019), characterization of combustion processes (Ahonen et al., 2017), monitoring occupational health

hazards (Gao et al., 2019) and clean room monitoring systems (Giechaskiel et al., 2009).

CPCs are based on two fundamental processes: heterogeneous nucleation and subsequent condensational growth. The seed particles are mixed with a condensable vapor, followed by a rapid temperature change leading to vapor supersaturation, particle activation by heterogeneous nucleation and subsquent condensational growth. The seed particles grow to sizes large enough where they can be detected optically. This process can be achieved by different working fluids such as n-butanol, water or DEG

which are commonly used in commercial instruments. Previous research has shown, that the molecular weight and the surface tension of the working fluid have an impact on particle activation (Iida et al., 2009; Magnusson et al., 2003). Higher molecular weights and higher surface tensions allow for the activation of smaller particles.

The detection performance of CPCs is commonly quantified by the 50 % cut-off diameter ($d_{p,50}$). The total detection efficiency of a CPC , $\eta_{tot}(d_p)$, is given by the following relation (Stolzenburg and McMurry, 1991):

$$\eta_{tot}(d_p) = \eta_s(d_p) \cdot \eta_a(d_p) \cdot \eta_d(d_p). \tag{1}$$

Here, $\eta_s(d_p)$ is the sampling efficiency accounting for the particle losses inside the instrument. $\eta_a(d_p)$ corresponds to the activation efficiency of the seed particles by heterogeneous nucleation and $\eta_d(d_p)$ is the detection efficiency of the particles in the optical system.

Heterogeneous nucleation processes are influenced by the physicochemical properties of the seed particles. If the seed particle

is entirely soluble in the condensing liquid, Köhler theory (Köhler, 1936) can describe the activation behaviour. If the seed is insoluble, heterogeneous nucleation theory (Fletcher, 1958) is applicable. Other relevant seed particle properties for their activation are size, shape, wettability and charging state (Kupc et al., 2013b). Previous studies have shown that the charging state of the seed particles might also have an influence on their respective detection efficiencies (Kangasluoma et al., 2016; Winkler et al., 2008). Chemical processes on the molecular level mediate particle activation, including interactions related to

the chemical composition of the seed particles and the molecules of the working fluids (Tauber et al., 2019a; Kangasluoma et al., 2016; Petäjä et al., 2006). Previous research has identified potential molecular characteristics that act as docking stations to the seed particle. These docking stations are polar groups, like -OH (Li and Hogan Jr., 2017). The distribution of polar groups, in general, defines the chemical polarity of a vapor molecule. A table summarizing relevant physical properties of the working fluids can be found in the Supplementary Information (SI, s. Table S1). Furthermore it has been shown that the





activation probability is influenced by the supersaturation occurring in the condenser of the CPC and the used working fluid (Iida et al., 2009; Wimmer et al., 2013). As a result the combination of CPCs based on different working fluids might even provide information concerning the chemical composition of the seed particles by showing differing detection efficiencies (Kulmala et al., 2007; Kangasluoma et al., 2014).

Here we present the results of studying the effect of the working fluid on the activation efficiencies with respect to various seed particle compositions. The broad array of investigated particle detectors was composed of n-butanol-based CPCs (TSI 3772, TSI 3776 and Airmodus A20), water-based CPCs (TSI 3788 and V-WCPC 3789) as well as a DEG-based booster stage (TSI 3777). The experiments have been performed under reproducible laboratory conditions and provide further insights into particle activation in CPCs.

## 10  2   Experimental Methods

A schematic of the experimental setup is displayed in Figure 1. The counting efficiencies of the CPCs were measured using four different types of seed particles, generated from sodium chloride (NaCl, pro analysi, Merck KGaA, Darmstadt, Germany), silver (Ag, wool for elemental analysis, CAS No.: 7440-22-4, Merck KGaA, Darmstadt, Germany), ammonium sulfate ($(NH_4)_2SO_4$, pro analysi, Merck KGaA, Darmstadt, Germany) and a solution of $\beta$-caryophyllene ($C_{15}H_{24}$, BCY, CAS No.:

87-44-5, Sigma Aldrich, St. Louis, USA). To avoid a possible influence of the relative humidity of the carrier gas on the detection efficiency (Tauber et al., 2019a; Kangasluoma et al., 2016), synthetic air (Alphagaz 1 Air, 99.999 % (5.0), $H_2O{<}3.0$ ppm$\cdot$mol$^{-1}$, Air Liquide) was used as a carrier gas and the aerosol was generated in dry conditions in all measurements. Sodium chloride, silver and ammonium sulfate particles were produced in a tube furnace (Scheibel and Porstendorfer, 1983), manufactured by Carbolite Gero GmbH & Co. KG, Germany. The particle material is put onto a crucible and inserted into the

cylindrical furnace. The material evaporates in the heated furnace and rapid cooling downstream of the heated section leads to particle formation. Table S2 in the SI contains the approximate furnace temperatures for different seed particle materials. Subsequent to the aerosol generator, a dilution flow of synthetic air is joined with the aerosol flow. A soft X-ray charger (TSI 3088 Advanced Aerosol Neutralizer) is used to achieve a steady-state charge distribution for the analyzed aerosol flow. In the next step, the aerosol enters a custom-made Vienna-type DMA (similar to Winklmayr et al. (1991) and referred to as nano

DMA), where particles are selected according to their electrical mobility. The electrical mobility diameters of the selected particles range between 1 and 25 nm. According to Flagan (1999), the flow rate ratio $\delta$ defines the limiting resolution in the absence of diffusion when the DMA flows are balanced and is given by the sum of the aerosol inlet flow $Q_a$ and the sample outlet flow $Q_s$ divided by the sum of the sheath flow rate $Q_{sh}$ and the excess flow rate $Q_e$:

$$\delta = \frac{Q_a + Q_s}{Q_{sh} + Q_e}. \tag{2}$$

The limiting resolution throughout our experiments is 0.15. By applying positive voltage to the nano DMA, particles with negative charge were selected. Negatively charged particles of the same composition are assumed to achieve higher activation effciencies at small sizes independently of the used CPCs (Kangasluoma et al., 2016; Winkler et al., 2008) and are thus


preferred for applications in size-distribution measurements down to particles as small as 1 nm. Previous studies on the counting efficiency of CPCs involving sub-2 nm particles were conducted using high resolution DMAs (Attoui and Kangasluoma, 2019). In order to assess the suitability of the used nano DMA for seed particles with diameters smaller than 2 nm, previous measurements of negatively charged Ag and NaCl seeds using a tuned TSI 3776, the TSI 3777 and a high resolution UDMA

(Steiner, 2011) were used. Parts of the results and the used experimental setup were published by Brilke et al. (2020). The results of the comparison are displayed in the SI.

A flow unit, including a total particle filter and a silica gel dryer, maintains a sheath air flow rate of 19.5 lpm (s. Fig.1a) or 33.0 lpm (s. Fig. 1b), respectively, in the DMA. The silica gel was exchanged frequently in order to maintain a dry sheath air cycle (closed-loop system). Downstream of the DMA the aerosol is evenly distributed among a CPC and a Faraday Cup

Electrometer (TSI 3068B Aerosol Electrometer) using a T-junction. The flow rates were adjusted and monitored using a TSI 4140 Mass Flow Meter. For the measurements with the DEG-based booster stage, a different flow setup was used due to the higher inlet flow rate of the instrument (s. Fig. 1b). Accordingly, the tube furnace was operated with 2.5 lpm of synthetic air and the aerosol inlet flow of the nano DMA was increased to 5.0 lpm. Consequently, the used setup for the generation of seed particles based on NaCl, Ag and $(NH_4)_2SO_4$ bears a close resemblance to the setup described in Kangasluoma et al. (2013),

where the authors also present mass spectra of furnace-generated clusters. The study by Kangasluoma et al. (2013) can be used to infer the chemical compostion of some of the seed particles used in this study.

Particles consisting of oxidized $\beta$-caryophyllene ($C_{15}H_{24}O_x$, $BCYO_x$) were produced in a flow tube (Hearn and Smith, 2006). The seeds were produced by evaporating a BCY solution into a clean airstream and subsequently mixing it with ozone, allowing the ozonolysis reaction to take place inside the flow tube. The mode diameter of the resulting size distribution can be shifted

by varying the length of the reaction path of the flow tube using an adjustable piston and thus the reaction time of the organic compounds. The diameter of the used flow tube was 0.05 m and the length of the reaction path was varied between 0.1 and 0.4 m. The temperature of the BCY evaporator was kept at 283.15 K and the ozone concentrations were adjusted to 100-500 ppb. The used CPCs and the booster stage are presented in Table 1. All particle counters are based on the laminar-flow principle and have 50 % cut-off diameters below 10 nm according to the manufacturers. Here, it should be noted that the TSI 3789

was operated in the 2 nm-mode for all experiments. The 2 nm-mode is the default setting by the manufacturer for low cut-off measurements based on calibrations with sucrose. The n-butanol and DEG-based instruments implement a typical architecture, consisting of a heated saturator and a cooled condenser. In case of the TSI 3772 and the Airmodus A20 the entire aerosol flow reaches the heated saturator. The TSI 3776 and TSI 3777 are additionally equipped with a capillary regulating the aerosol flow (Stolzenburg and McMurry, 1991). This capillary-sheath layout helps to keep the aerosol flow centered and enhances the

detection efficiency by achieving a higher and sharper supersaturation profile within the condenser and reducing diffusional losses (Stolzenburg and McMurry, 1991). In the case of the water-based TSI 3788 a cooler conditioner is followed by a heated growth tube (Hering et al., 2005). Here, the aerosol flow is first cooled and then enters a heated region of supersaturated vapor. The temperatures of the sections have to be switched due to the higher diffusion coefficient of water compared to heat transfer. Lastly, the TSI 3789 (V-WCPC 3789) is based on a three-step principle: a cool conditioner, a warm initiator and a

cooler moderator. The moderator is necessary for removing water vapor and heat while maintaining supersaturated conditions



(Hering et al., 2019). In contrast to the TSI 3788, the TSI 3789 is not based on a capillary-sheath layout.

By changing the internal settings of three selected CPCs, the 50 % cut-off diameters were pushed to smaller sizes using BCYO$_x$ particles. In case of the TSI 3777 and the TSI 3789, the boosting was achieved by changing the temperature settings. In case of the tuned TSI 3776 (denoted as TSI 3776*), the internal flows were also adjusted. The valve regulating the sheath air flow

is adjusted such, that the sample flow rate of the CPC increases to 2.5 lpm (Barmpounis et al., 2017; Brilke et al., 2020).

The detection efficiency $\eta$ is determined by comparing the particle number concentration measured by a CPC, $N_{CPC}$, to the particle number concentration of the FCE, $N_{FCE}$:

$$\eta = \frac{N_{CPC}}{N_{FCE}}. \tag{3}$$

After the tube furnace had reached a stable state, the number concentrations were simultaneously measured for 45 seconds.

Uncertainty analysis of measurement data was performed according to the rules of Gaussian error propagation.

## 3    Results and Discussion

### 3.1    Composition Dependent Counting Efficiencies

Figure 2 shows the detection efficiencies measured with two CPCs (TSI 3776 and TSI 3789) and the CPC-conditioner combination (TSI 3777 and TSI 3772). The abscissa is kept between 1 and 10 nm in order to set focus on the differences between

the used seed particles. The detection efficiency shows a clear dependence on the seed particle composition and the working fluid. The smallest 50 % cut-off diameters using the DEG-based TSI 3777 were accomplished by using NaCl and $(NH_4)_2SO_4$ seeds. Both seed particles showed a 50 % cut-off diameter of $(1.6\pm0.2)$ nm. In case of the TSI 3776, BCYO$_x$ seeds were found to have the smallest 50 % cut-off diameter of $(2.8\pm0.3)$ nm. Lastly, NaCl and $(NH_4)_2SO_4$ seeds exhibited the smallest 50 % cut-off diameter using the TSI 3789 ($(2.3\pm0.1)$ nm). Interestingly, the TSI 3777 shows a significant difference between the

activation of ionically bond (and polar) salts (NaCl and $(NH_4)_2SO_4$) and less polar seed particles (Ag and BCYO$_x$): The polar compounds are activated at smaller diameters compared to the nonpolar ones. In case of the TSI 3776 the detection efficiency curves are very similar for all seed particles, except for NaCl. The related detection efficiency curve is shifted towards larger diameters. The water-based CPC shows a much smaller 50 % cut-off diameter for NaCl and ammonium sulfate compared to the butanol-based counter ($(2.30\pm0.12)$ nm vs. $(4.08\pm0.51)$ nm for NaCl). Overall, the difference between the smallest and

the largest 50 % cut-off diameter is approximately 1 nm for all three CPCs.

The measured detection efficiency curves show the following two features: (i) The slopes corresponding to the DEG-based TSI 3777 are steeper compared to the n-buatnol-based TSI 3776 and the water-based TSI 3789. (ii) The counting efficiency of the TSI 3776 and the TSI 3777 reaches 1 around 10 nm in both cases. On the contrary, the detection efficiency of the TSI 3789 saturates around 1 at about 20 nm (s. Fig. S2 and S3 in the SI).





### 3.2 Comparison of the Water-Based CPCs

Subsequently, the two investigated water-based CPCs were directly compared to each other. Figure 3 shows a direct comparison of the detection efficiency curves for the two different water-based CPCs (TSI 3788 and TSI 3789). The TSI 3788 shows almost identical curves for NaCl and ammonium sulfate with a 50 % cut-off diameter of (2.2±0.1) nm for ammonium sulfate, while

the curves for the less polar compounds (BCYO$_x$ and Ag) are shifted by approximately 1 nm (s. Fig. 3a). The TSI 3789, on the other hand, shows a different kind of compound dependence, since the detection efficiency curves for polar and less polar seeds are not separated as clearly. All 50 % cut-off diameters range from 2.3-3.4 nm and are similar to the cut-off diameters of the TSI 3788, with NaCl having the lowest $d_{p,50}$ and BCYO$_x$ having a rather high $d_{p,50}$. The curves measured with the TSI 3788 are steeper compared to the TSI 3789. As shown in Fig. 3c, the lower detection efficiencies for the TSI 3789 below 10

nm are due to to higher internal losses compared to the TSI 3788. This behaviour is most probably linked to the fact that the TSI 3789 is not based on a capillary-sheath layout. The upper curve shows the detection efficiency corrected for diffusional losses in the inlet and conditioner according to Gormley and Kennedy (1948) for Ag seeds. The corrected detection efficiency curves of the other seeds are presented in the SI (s. Fig. S4).

### 3.3 Comparison of the N-Butanol-Based CPCs

Figure 4 shows the direct comparison of the butanol-based CPCs. Three out of the four butanol-based CPCs, i.e. the TSI 3772, the TSI 3776 and the Airmodus A20, show the expected activation pattern: BCYO$_x$ and Ag seeds are activated at smaller diameters compared to $(NH_4)_2SO_4$ and NaCl seeds. The Airmodus A20 reaches its plateau, located at detection efficiency values of about 0.9 at 20 nm.

Astoundingly, the TSI 3776* shows barely no composition dependence of the 50 % cut-off diameters, thereby confirming the

results for of Brilke et al. (2020) for Ag seeds. There is no significant difference between the smallest and the largest 50 % cut-off diameter, which are centered around 2.1 nm. The detection efficiency curves are steep and reach plateau levels of 1 at already 4.5 nm.

### 3.4 50 % Cut-Off Diameters

Figure 5 summarizes the 50 % cut-off diameters measured with the aforementioned instruments operated at standard settings.

The discussed dependence of the detection efficiencies on the working fluids and the seed particle compositions can be seen clearly. The DEG-based TSI 3777 and the two water-based CPCs (TSI 3788 and TSI 3789) show the smallest 50 % cut-off diameters for polar NaCl and ammonium sulfate seeds. The n-butanol-based CPCs activate less polar BCYO$_x$ seeds at the smallest 50 % cut-off diameters. Additionally, our results clearly show that the dependence of the detection efficiency on the seed particle material also influences CPCs with 50 % cut-off diameters larger than 5 nm as the absolute shifts in the diameters get larger for these models. This applies for the Airmodus A20 and the TSI 3772.





### 3.5 Effect of Instrument Boosting

As the internal temperature settings in all CPCs used in this work can be adjusted by the customer, a set of experiments was performed to test the instruments with different temperature settings (s. Fig. 6). The tests were done using BCYO$_x$ seed particles and the tested CPCs were the butanol-based TSI 3776, the DEG-based TSI 3777 and the water-based TSI 3789. In

case of the TSI 3777 and the TSI 3789, only the temperatures were adjusted and for the TSI 3776 also the flows were changed (TSI 3776*, s. Table 2 for tuned settings). The adjustment of the temperatures did not increase the background count rate of the instruments. The results are shown in Figure 6. For all tested CPCs, the 50 % cut-off diameters could be reduced (s. Table 3). The shapes of the curves did not show any significant change except for the TSI 3776*, which is due to a change of the internal flow rates, leading to reduced diffusion losses at the instrument's inlet. As a result, the overall counting efficiency is

enhanced and exhibits a steeper slope. The 50 % cut-off diameter could be lowered by approximately 35 % in case of the TSI 3776, the TSI 3777 and the TSI 3789. The effect of just readjusting temperatures can be clearly seen too by comparing Fig. 6a and Fig. 6b.

### 3.6 Discussion of the Measurement Results

The results of the performed measurements show that chemical similarities between the seed particle material and the working

fluids influence the detection efficiencies of CPCs. As a rule of thumb, compounds of similar chemical structure dissolve more easily. N-butanol is a rather non-polar fluid and interacts stronger with particles of non-polar substances. On the contrary, water and DEG are highly polar fluids, and ammonium sulfate and NaCl are ionic compounds. Therefore it is expected, that ionic compounds such as NaCl and ammonium sulfate easily dissolve in the polar working fluids and that the less polar BCYO$_x$ and Ag particles interact stronger with the non-polar n-butanol. These chemical similarities of seed and working fluid are

indeed reflected in the counting efficiencies: The 50 % cut-off diameters of the ionic seeds are smaller when measured with the TSI 3789 or TSI 3777. On the other hand, the 50 % cut-off diameters of Ag and BCYO$_x$ seeds are smaller when measured with a butanol-based CPC (s. Fig. 2-Fig. 4). This different activation behaviour also requires different theoretical approaches: While particle activation of two soluble compounds can be described by Köhler theory (Köhler, 1936), Fletcher theory of hetergeneous nucleation (Fletcher, 1958) needs to be applied when the seed particle is not entirely soluble in the working fluid

(Giechaskiel et al., 2011).

Additionally, the detection efficiency of the instruments is influenced by technical factors. The TSI 3776 and the TSI 3777 are based on a capillary-sheath layout. As a result, particle losses due to diffusion are smaller and the detection efficiencies reach higher values at smaller diameters compared to the TSI 3789 (s. Fig 2). Additionally, the detection efficiency curves become steeper.

By changing the temperature settings, higher peak saturation ratios are achieved, leading to smaller 50 % cut-off diameters (s. Fig. 6). The chosen settings increase the temperature difference between the cooled and heated regions inside the instruments, while simultaneously avoiding significant homogeneous nucleation. Homogeneous nucleation inside the instrument is tested by connecting a total particle filter to the inlet and verifying that no signal is detected. As shown by Tauber et al. (2019b),

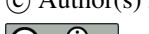



lowering the operating temperatures of a TSI 3776 by keeping the temperature difference between saturator and condenser constant leads to an elevated saturation ratio profile in the condenser tube.

Tauber et al. (2019a) presented results of detection efficiency measurements of the TSI 3776 using Ag and NaCl seeds. Our results for Ag and NaCl are in perfect agreement with the results for a dry aerosol flow of Tauber et al. (2019a). As Brilke et al.

(2020) highlight, there is no composition dependence of the detection efficiencies of the TSI 3776*. The 50 % cut-off diameters for negatively-charged Ag seeds presented by the authors are also confirmed by our measurements. We report the same 50 % cut-off diameters for the TSI 3776* and $(NH_4)_2SO_4$ as reported by Kangasluoma et al. (2016), who also used a tuned TSI 3776. We confirm the discussed trend of smaller 50 % cut-off diameters when using CPCs based on water and DEG (with DEG being linked to the smallest one) is backed by our results. Kupc et al. (2013a) performed a laboratory characterization of

the TSI 3788 and compared the results to the TSI 3776. The used seed particles also include negatively-charged NaCl seeds. The 50 % cut-off diameters for positively-charged NaCl measured with the TSI 3788 are in good agreement, but we report distinctly larger 50 % cut-off diameters for positively-charged NaCl measured with the TSI 3776. In their publication about DEG-based particle counters Wimmer et al. (2013) present 50 % cut-off diameters for negatively-charged $(NH_4)_2SO_4$ seeds. The results of our study also support these measurements.

### 3.7 Supersaturation Profiles

To confirm our measurement results regarding the cut-off diameters with theoretical calculations, we simulated the maximum supersaturations for the three butanol-based CPCs (TSI 3776, TSI 3776* and the TSI 3772). The supersaturations in the Airmodus A20 were provided by the manufacturer. Measurements of neutral Ag seeds performed with the Size Analyizing Nuclei

Counter (SANC, Tauber et al. (2019a); Wagner (1985)) were used to correctly simulate the maximum supersaturation seed particles were exposed to (Tauber et al., 2019b). Therefore, we evaluated the heat and mass transfer for a fully developed laminar flow with no mixing, according to Stolzenburg and McMurry (1991). The original geometry was transformed to a circular tube to solve the equations with constant boundary conditions (constant temperatures, Eckert and Drake (1972)). For the flow a parabolic velocity flow profile was considered and used to simulate the convective/diffusive heat and mass transfer for the

simulation domain following Tauber et al. (2019b). The results are presented in Table 4, showing the calculated corresponding minimum diameters that can be activated. The highest supersaturations ($S_{max}$=4.61) are reached in the TSI 3776* and particles are exposed to the lowest supersaturations in the TSI 3772 ($S_{max}$=1.93), as expected, according to our calculations.

We correlated the measured 5 % cut-off diameter with the diameters that can be theoretically activated ($d_{p,0}^T$). The comparison was done for silver particles. In the theoretical calculations neutral particles were used, whereas in the case of the

measurements we used negatively charged silver particles. Figure 7 shows a remarkable correlation between the theory and the measurements, resulting in an $R^2$ of 0.98. In the case of the smallest diameter, the calculated diameter is 1.8 nm and the measured one is (1.5±0.2) nm. This deviation is most likely due to the charge effect (Tauber et al., 2018). We are therefore able to correctly predict the onset of detection in CPCs from simulated supersaturation profiles, which implies that we can



infer the maximum supersaturation inside the condenser for specific CPCs, which was generally unknown for most CPCs so far. Calculated supersaturation profiles can be found in the SI (s. Fig. S5 and Fig. S6).

## 4    Conclusions

The effect of the chemical composition of seed particles on the detection efficiency in commonly used CPCs based on different working fluids was investigated. These characterizations included organic seed particles, that were generated in a controlled way. We present the first characterization measurements of the newly developed TSI 3789 using seed particles with diameters between 1 and 25 nm. Furthermore, the onset of detection in CPCs was successfully and correctly predicted based on simulated supersaturation profiles.

Chemical similarities between the seed particle material and the used working fluids have been found to have an impact on the detection efficiency of CPCs. It has been shown, that shifts in the detection efficiencies also occur for CPCs with 50 % cut-off diameters larger than 5 nm. Additionally, it was confirmed that the detection efficiency can be improved by changing the temperature and flow rate settings of the instruments. Remarkably, the TSI 3776 did not show any shifts in the detection efficiency also for NaCl, $(NH_4)_2SO_4$ and $BCYO_x$ seeds, when operated using the tuned settings. This behaviour is linked to the higher saturation ratio after tuning the CPC. The saturation ratio might be high enough that no differences in the detection efficiencies of different seed particles occur anymore.

Consequently, we conclude that the 50 % cut-off diameter as sole parameter is not sufficient in characterizing the detection efficiency of a CPC. Shifts of the detection efficiency and curve shapes are influenced by the aforementioned interactions. We recommend that CPC characterizations should be performed using various polar and less polar seed particles, including Ag seeds, in order to correctly present the detection efficiency of a CPC. The chemical composition of the measured aerosol particles should be considered when instruments are calibrated. The authors recommend to follow the calibration standard based on Ag seeds, introduced by Wiedensohler et al. (2018), when a variety of seed particle materials is not at hand. In the future, measurements of the activation efficiency with Ag could then be used to infer the achieved supersaturation in the CPC. Subsequent comparison to further SANC measurements with other seed materials (e.g. NaCl, Tauber et al. (2019b)) than allow to infer the corresponding counting efficiencies. Hence, this study suggests a novel approach in determining counting efficiencies of CPCs when a calibration with a variety of seed-particles are not available.

*Data availability.* Raw data are available upon request from the authors.

*Author contributions.* DW, SHS and PMW presented the idea. PJW, DS, SB and DW designed the setup and performed the measurements. CT calculated the supersaturation profiles for the TSI CPCs. PJW, DS, CT and DW analyzed the data. PJW, DS, CT, SB, SHS, PMW and DW were involved in the scientific interpretation of the results. PJW, DS, SB and DW wrote the manuscript. All authors participated in reviewing the manuscript.





*Competing interests.* All other authors declare that they have no conflict of interest.

*Acknowledgements.* The authors thank TSI Inc. for kindly providing a TSI 3789 for the measurements. Furthermore the authors acknowledge A. Zerrath for providing technical support concerning the newly-developed TSI 3789. J. Enroth is acknowledged for calculating the supersaturation profile of the Airmodus A20 and providing it to the authors.

5   This work was supported by the Austrian Science Fund (FWF) Project J3951-N36 and the European Research Council (ERC) Consolidator Grant NANODYNAMITE 616075. This study was independently performed and was not co-funded by TSI.



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



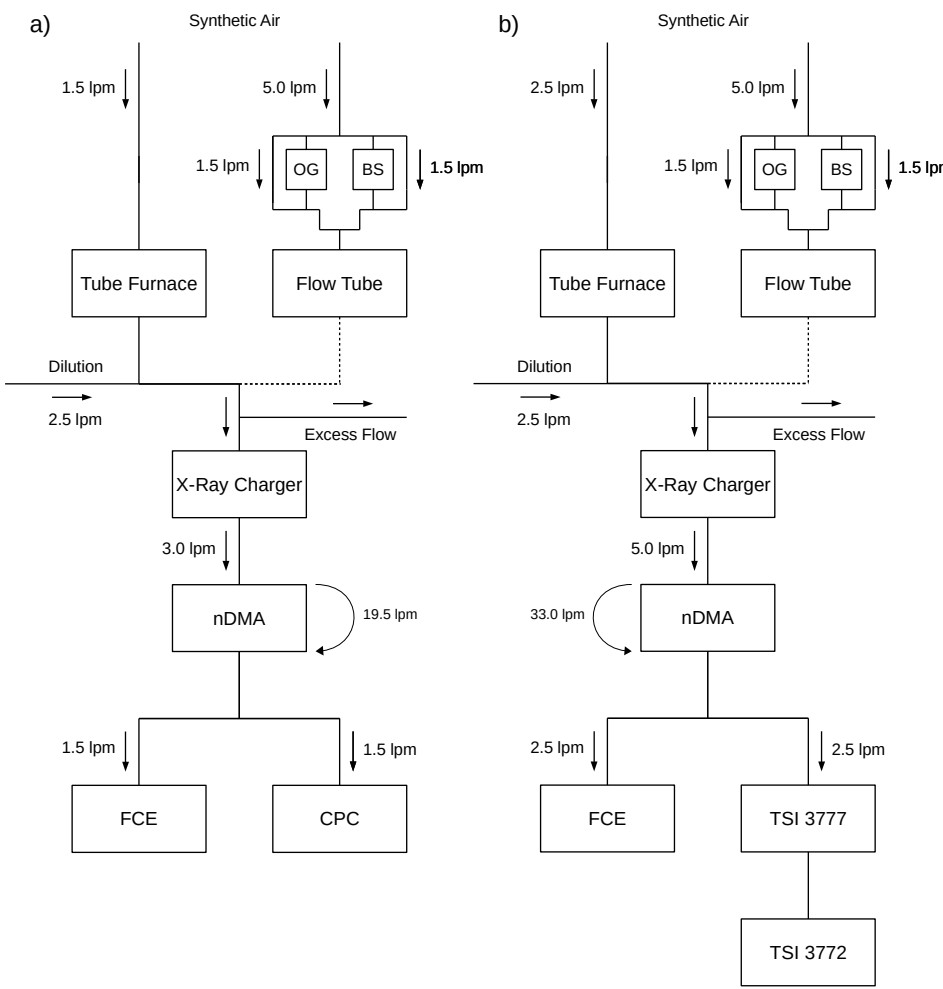

**Figure 1.** Schematic of the Experimental Setup: The Figure shows a schematic of the experimental setup that was used to measure the detection efficiency of the ultrafine particle counters. A Faraday Cup Electrometer was used as a reference. $C_{15}H_{24}O_x$ seeds were generated using a BCY solution (BS), an ozone generator (OG) and a flow tube. The curved arrows mark the sheath air cycle. Panel a shows the setup for the measurements involving just CPCs. Due to the higher aerosol inlet flow rate of the TSI 3777 the setup was modified (Panel b).





**Table 1.** Used Instrumentation: The table shows the used particle counters, including their model numbers, the used working fluids as well as the 50 % cut-off diameter as stated by the manufacturers.

| Manufacturer | Model | Working Fluid | $d_{p,50}$ [nm] |
|---|---|---|---|
| TSI | 3776 | n-butanol | 2.5 |
| TSI | 3772 | n-butanol | 10.0 |
| TSI | 3788 | water | 2.5 |
| TSI | 3789 | water | 2.2 |
| TSI | 3777 | DEG | 1.4 |
| Airmodus | A20 | n-butanol | 7.0 |





**Table 2.** Temperature Settings: The table summarizes the standard temperature settings according to the user manuals of the particle counters as well as empirically derived tuned temperature settings (s. also Brilke et al. (2020)).

| Standard T-Settings [°C] | | | | | |
|---|---|---|---|---|---|
| Instrument | $T_{Condenser}$ | $T_{Saturator}$ | $T_{Optics}$ | $T_{Conditioner}$ | $T_{Initiator}$ | $T_{Cabinet}$ |
| TSI 3776 | 10.0 | 39.0 | 40.0 | - | - | - |
| TSI 3789 | - | - | 40.0 | 7.0 | 90.0 | 15.0 |
| TSI 3777 | 12.0 | 62.0 | - | - | - | - |

| Tuned T-Settings [°C] | | | | | |
|---|---|---|---|---|---|
| Instrument | $T_{Condenser}$ | $T_{Saturator}$ | $T_{Optics}$ | $T_{Conditioner}$ | $T_{Initiator}$ | $T_{Cabinet}$ |
| TSI 3776* | 1.1 | 33.1 | 34.1 | - | - | - |
| TSI 3789* | - | - | 40.0 | 2.0 | 95.0 | 23.0 |
| TSI 3777* | 12.0 | 68.0 | - | - | - | - |



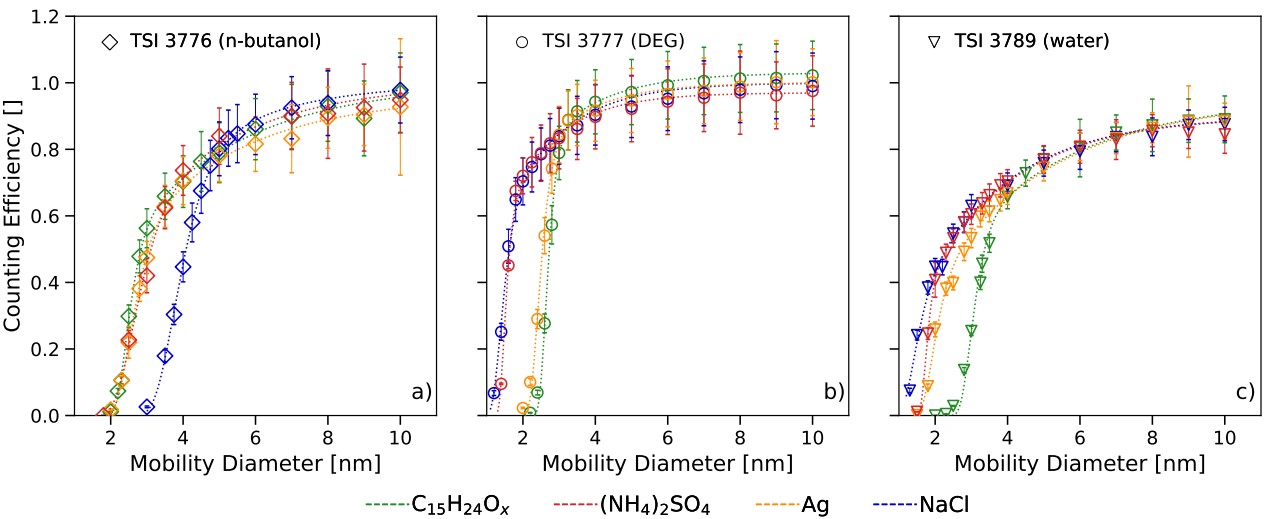

**Figure 2.** Detection Efficiencies and Working Fluids: The Figure shows the detection efficiencies for different seed particle materials as a function of the electrical mobility equivalent diameter. Different colors correspond to different seed particles and every plot is related to a different working fluid: n-butanol (Panel a), DEG (Panel b) and water (Panel c).



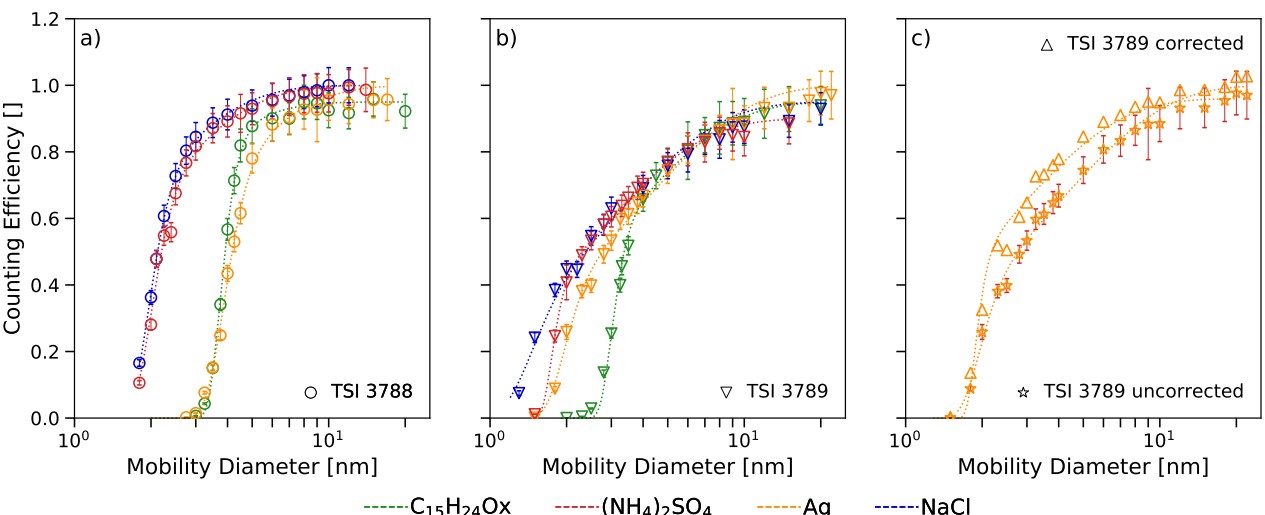

**Figure 3.** Detection Efficiencies of the Water-Based CPCs: The Figure shows the detection efficiencies as a function of the electrical mobility equivalent diameter. Panel a corresponds to the TSI 3788 and Panel b shows the data measured with the TSI 3789. Panel c displays the data set for Ag that has been corrected for diffusion losses; Panel a and Panel b show uncorrected data. The abscissa is scaled logarithmically to set focus on the dependence of the detection efficiencies on the seed particle material.

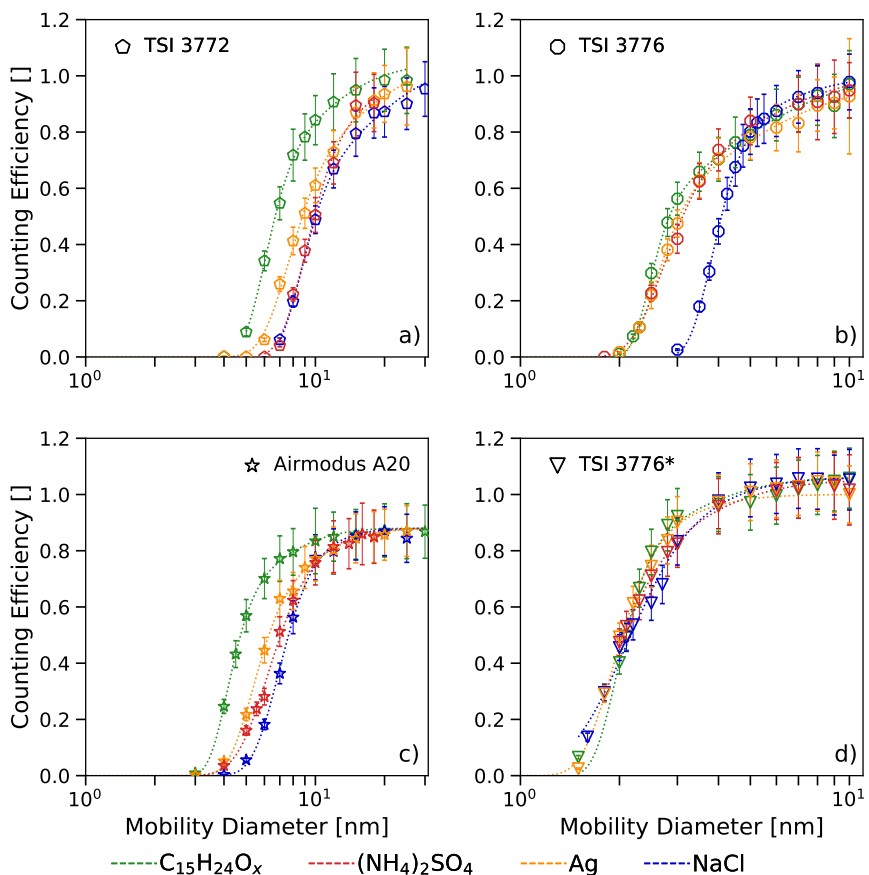

**Figure 4.** Detection Efficiencies of the Butanol-Based CPCs: The Figure shows the detection efficiencies as a function of the electrical mobility equivalent diameter. Every plot corresponds to a different butanol-based CPC: TSI 3772 (Panel a), TSI 3776 (Panel b), Airmodus A20 (Panel c) and TSI 3776* (Panel d). The abscissa is scaled logarithmically to set focus on the dependence of the detection efficiencies on the seed particle material.

Atmospheric
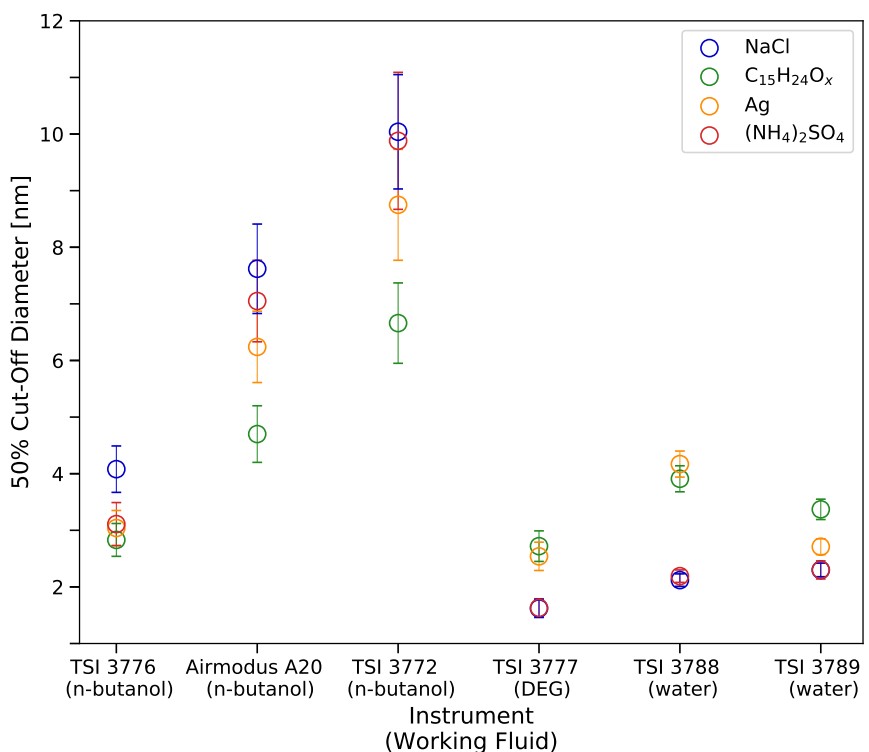

**Figure 5.** 50 % Cut-off Diameters for Different Working Fluids: The Figure shows the 50 % cut-off diameters measured with the TSI 3776, the Airmodus A20, the TSI 3772, the TSI 3777, the TSI 3788 and the TSI 3789. Different colors corresponds to different seed particle compositions.



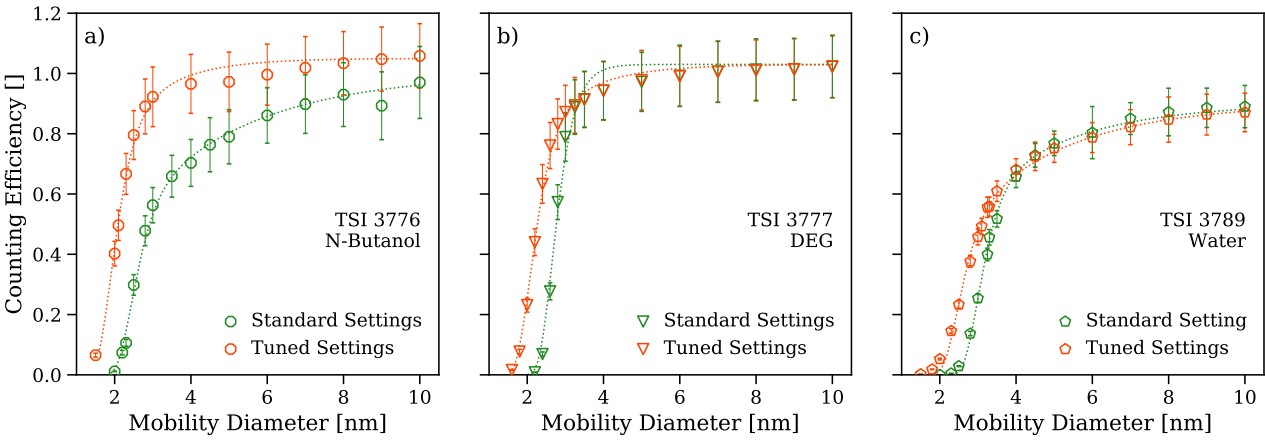

**Figure 6.** Detection Efficiencies of the Tuned Instruments: The Figure shows the detection efficiencies for $BCYO_x$ seeds as a function of the electrical mobility equivalent diameter. The green lines correspond to the standard settings, the orange ones to the tuned settings. Every plot shows data of a different instrument: TSI 3776 (Panel a), TSI 3777 (Panel b) and TSI 3789 (Panel c).



**Table 3.** Summarized 50 % Cut-Off Diameters: The table shows the measured 50 % cut-off diameters for the four different seed particle materials and for every particle counter in use.

| Seed | Working Fluid | Instrument | $d_{p,50}$ [nm] | $\Delta d_{p,50}$ [nm] |
|---|---|---|---|---|
| NaCl | n-butanol | TSI 3776 | 4.0 | 0.4 |
|  |  | TSI 3776* | 2.2 | 0.2 |
|  |  | TSI 3772 | 10.0 | 1.0 |
|  |  | A20 | 7.6 | 0.8 |
| NaCl | water | TSI 3788 | 2.1 | 0.1 |
|  |  | TSI 3789 | 2.3 | 0.1 |
| NaCl | DEG | TSI 3777 | 1.6 | 0.2 |
| Ag | n-butanol | TSI 3776 | 3.0 | 0.3 |
|  |  | TSI 3776* | 2.0 | 0.2 |
|  |  | TSI 3772 | 8.8 | 1.0 |
|  |  | A20 | 6.2 | 0.6 |
| Ag | water | TSI 3788 | 4.2 | 0.2 |
|  |  | TSI 3789 | 2.7 | 0.1 |
| Ag | DEG | TSI 3777 | 2.5 | 0.3 |
| $(NH_4)_2SO_4$ | n-butanol | TSI 3776 | 3.1 | 0.4 |
|  |  | TSI 3776* | 2.0 | 0.2 |
|  |  | TSI 3772 | 9.9 | 1.2 |
|  |  | A20 | 7.1 | 0.7 |
| $(NH_4)_2SO_4$ | water | TSI 3788 | 2.2 | 0.1 |
|  |  | TSI 3789 | 2.3 | 0.2 |
| $(NH_4)_2SO_4$ | DEG | TSI 3777 | 1.6 | 0.2 |
| $C_{15}H_{24}O_x$ | n-butanol | TSI 3776 | 2.8 | 0.3 |
|  |  | TSI 3776* | 2.1 | 0.2 |
|  |  | TSI 3772 | 6.7 | 0.7 |
|  |  | A20 | 4.7 | 0.5 |
| $C_{15}H_{24}O_x$ | water | TSI 3788 | 3.9 | 0.2 |
|  |  | TSI 3789 | 3.4 | 0.2 |
|  |  | TSI 3789* | 3.1 | 0.2 |
| $C_{15}H_{24}O_x$ | DEG | TSI 3777 | 2.7 | 0.3 |
|  |  | TSI 3777* | 2.3 | 0.2 |





**Table 4.** Calculated Supersaturation Ratios: The table shows the results of the modelling of the supersaturation profiles of four different particle counters. The maximum supersaturation ratio is related to the smallest particle diameter necessary for the activation of neutral silver seeds.

| Instrument | $S_{max}$ [] | $d_{p,0}$ [nm] |
|---|---|---|
| TSI 3776 | 3.43 | 2.20 |
| TSI 3776* | 4.61 | 1.80 |
| TSI 3772 | 1.68 | 5.40 |
| Airmodus A20 | 1.93 | 4.20 |

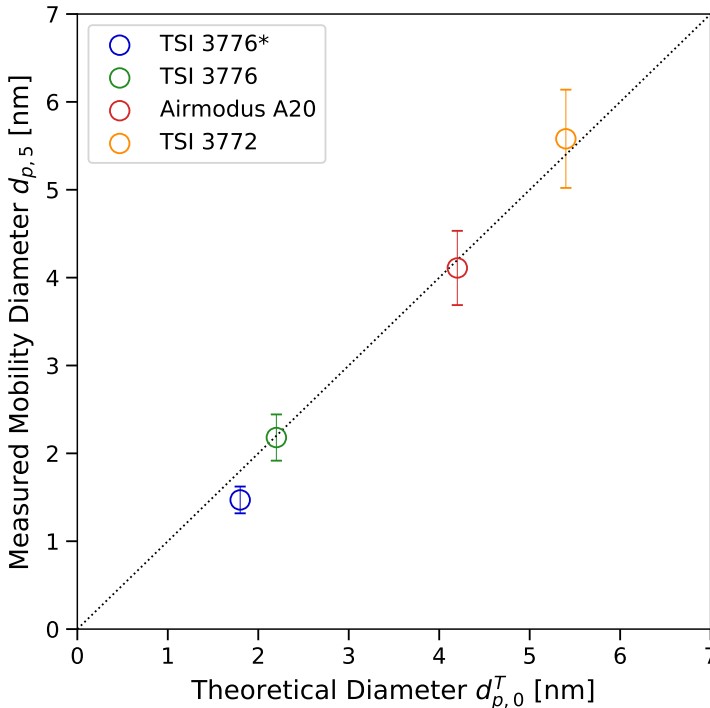

**Figure 7.** Theoretical Diameters: The Figure shows the theoretically minimal diameters necessary for particle activation of neutral silver seeds as a function of the measured 5 % cut-off diameter for negatively charged silver seeds. Every color corresponds to a different particle counter. Data points on the dashed line stand for the equality of the theoretical and measured diameters.