# Peer review of "Counting on Chemistry: Laboratory Evaluation of Seed Material-Dependent Detection Efficiencies of Ultrafine Condensation Particle Counters"

_Atmospheric Measurement Techniques, 2019_

## Referee Comment (RC1) · Anonymous Referee #3 · 3 Mar 2020

This paper investigates the effect of particle composition on the measured cutoff of a number of different CPC models. This is an important technical study, as this is (by default) usually characterised using silver nanoparticles, which may not be representative of the particles of interest in the atmosphere during new particle formation events. This paper highlights fundamental differences in the behaviour of different CPCs according to working fluids and whether the particles are organic or inorganic. While the results are not unexpected, they should assist in the characterisation of instruments and interpretation of ambient and laboratory data. This paper is within scope of AMT, the methods used are appropriate and the work methodically presented. I therefore recommend publication subject to the following technical comments:

General: Please use a different method to denote the tuned instrument rather than an asterisk, as this normally implies a footnote. Suggest superscript-'T'

Page 3, line 12: The purities and grades of all of the chemical stocks should be stated, including the solvent used for the BCY solution. Also state the concentration of the BCY solution.

Page 4, line 18: State the method used to generate ozone and control the concentration

Page 5, line 24: Please do not use the word 'saturates', as this could cause confusion.

Page 6, line 5: Is this not related to the particle's solubility rather than polarity?

Page 6, line 19: The word 'astoundingly' isn't particularly scientific. Please describe what aspect was unusual or unexpected.

Page 7, line 11: The phrase "The effect of just readjusting temperatures can be clearly seen too by. . ." is very clumsy. Please reword.

Page 8, line 32: Remove brackets around the reported statistic

Page 10, line 6: The statement about the work being performed independently should really come under the competing interests statement. Could the same statement about TSI be extended to Airmodus?

---

## Referee Comment (RC2) · Anonymous Referee #2 · 15 Mar 2020

I have two deep points of conflict with the authors of this work. These points are about particles smaller than 2 nm. The first point is about the particles themselves and the second point is about they are produced and selected. 1°/ The authors argue that a 11 cm Vienna DMA at 30 lpm sheath air can select 1 nm particles with an acceptable resolution. I guess that they have used the DMA 1/40 introduced by Georg Reischl (Winklmayr et al. 1991). ██████████████████████████████████ 2°/ The colleagues are using an X ray charger (to produce ions) for the generated particles before the DMA. This point is critical in my opinion. In the sub 2 nm a charger should be useless and even forbidden when the purity (chemistry in fact) is important. Indeed the figure 2 b, 2 c, 3b, 3c and 4d show results with particles smaller than 2 nm of silver

and sodium chloride for example. It is clear that these particles are not pure but dirt. Indeed if you add an ion to a particle to charge it in positive or in negative mode the result has nothing to do with the particle nor with the ion. Ag + Nitrate ion dimer (for example) from the charger is not pure silver particle anymore. Same thing with sodium chloride NaCl + lactic acid (for example) from has nothing to do with sodium chloride. See Maisser et al. (2015) JAS 90, 36-50; Steiner et al. (2014) AST 48,3 261-270 for the chemical composition of ions produced in chargers. I would suggest to the authors to be careful and warn the readers concerning the sub 2 nm results with a charger. High resolution DMAs, electrospray source atomization and evaporation condensation of vapors from hot wires are for the moment the cleanest methods for the generation of clean sub 2 nm. Clean means pure chemistry. Indeed the wire generator is much cleaner than an oven because the hottest point is the wire itself. It's not the case inside an oven. The particles from the wire generator are on the other hand self-charged. It's not the case when an oven is used.

I have few other small details about the work. 1°/ The following previous works should appear in the introduction of the paper to my opinion Seto et al (1997) https://doi.org/10.1063/1.474510 Gamero & Fernandez de la Mora https://doi.org/10.1016/S0021-8502(99)00555-8 Attoui 2018 : https://doi.org/10.1016/j.jaerosci.2018.08.005

2°/ I don't understand very well why the authors are using two identical set ups. What is the benefit of the DMA working at 19.5 lpm? Why not 20 lpm by the way? 3°/ It will be good to give the geometrical parameters of the DMAs. 4°/ The authors are giving the resolution of their DMA but nothing about the flowrate nor the size of the particles they have used for the measurement of the resolution. Was it done with a tandem DMA by the way? 5°/ The equation 2 is useless since the authors are talking about particles down to 1 nm (as small as !) where the diffusion is very active adversely to what Rick Flagan was telling in the cited paper. 6°/ In the figure 2 the diameter is given in standard number format. It's not the case in the figures 3 and 4. Please use

the same format. I would suggest standard format rather than scientific format. 7°/ Susan Hering and colleagues as published in a paper the experimental results of the versatile CPC commercialized by TSI. That paper is the most adequate reference to the instrument. The paper is cited on the TSI website. Susanne V. Hering, Gregory S. Lewis, Steven R. Spielman, Arantzazu Eiguren-Fernandez, Nathan M. Kreisberg, Chongai Kuang & Michel Attoui (2017) Detection near 1-nm with a laminar-flow, water-based condensation particle counter, Aerosol Science and Technology, 51:3, 354-362.

---

## Author Comment (AC1) · 11 Apr 2020

**Author's Response to Referee #3**

We thank Referee #3 for the critical assessment of our work and the helpful comments. In the following we address the comments point by point:

*General: Please use a different method to denote the tuned instrument rather than an asterisk, as this normally implies a footnote. Suggest superscript-'T'*

The suggestion will be implemented as stated. All relevant asterisks will be replaced by T's (Page/Line: 5/4, 6/19, 7/6, 7/8, 8/5, 8/7, 8/18, 8/26, Fig. 4, Table 3, Table 4 and Fig. 7)

*Page 3, line 12: The purities and grades of all of the chemical stocks should be stated, including the solvent used for the BCY solution. Also state the concentration of the BCY solution.*

The purity of the used solvent for the BCY solution (water) as well as the concentration of the solution will be added (3/14 and 3/15).

*Page 4, line 18: State the method used to generate ozone and control the concentration*

An UV lamp was used to generate ozone. The concentrations were not monitored due to previous experiments on the performance of the lamp. The following sentence will be added:

4/19: Ozone was generated using a custom made UV-lamp with adjustable intensity. An intensity/ozone calibration was performed prior to the experiments with an ozone monitor (ThemoFischer Scientific Model i49), suggesting that the ozone concentrations were in the range of 100-500 ppb.

*Page 5, line 24: Please do not use the word 'saturates', as this could cause confusion.*

In order to avoid confusion, the word 'saturates' will be replaced by the word 'reaches'. The following changes will be made:

5/29: On the contrary, the detection efficiency of the TSI 3789 reaches 1 at

about 20 nm.

*Page 6, line 5: Is this not related to the particle's solubility rather than polarity?*

We suspect that especially in the sub-10 nm size range charge effects might play a crucial role during the activation of seed particles (s. Figure 7 and 8/32). In order to properly include that into our argumentation, we refrained from using the word 'solubility'. Nevertheless, the observed effects are very similar to dissolution processes.

*Page 6, line 19: The word 'astoundingly' isn't particularly scientific. Please describe what aspect was unusual or unexpected.*

Since the observation is discussed in the following lines, the word 'astoundingly' will be deleted (6/19).

*Page 7, line 11: The phrase "The effect of just readjusting temperatures can be clearly seen too by. . ." is very clumsy. Please reword.*

The following changes will be made:

7/11: Fig. 6a shows the effect of changing the temperature settings and the inlet flow of the TSI 3776. The effect of just changing the temperature settings is shown in Fig. 6b and Fig. 6c (TSI 3777 and TSI 3789).

*Page 8, line 32: Remove brackets around the reported statistic*

The brackets will be removed throughout the text (5/17, 5/18, 5/19, 5/24, 6/4 and 8/32).

*Page 10, line 6: The statement about the work being performed independently should really come under the competing interests statement. Could the same statement about TSI be extended to Airmodus?*

The presented work has been performed without funding from any company. The statement is true for TSI Inc as well as Airmodus Ltd. The related statement will be moved into the "Competing Interest" - Section and Airmodus Inc. will be included.

The following changes will be made:

10/1: Competing interest: All authors declare that they have no conflict of interest. This study was independently performed and was not co-funded by TSI Inc and Airmodus Ltd.

---

## Author Response (AR2)

**Final Author's Response: Referee #2**

By P. J. Wlasits

peter.wlasits@univie.ac.at

We thank Referee #2 for the critical assessment of our work and the helpful comments. In the following we address the comments point by point:

*I have two deep points of conflict with the authors of this work. These points are about particles smaller than 2 nm. The first point is about the particles themselves and the second point is about they are produced and selected.*

In general, the aim of the presented study was to point out the importance of seed particle materials when it comes to calibrating condensation particle counters and their application for atmospheric measurements. Therefore our experiments were performed with instrumentation commonly used for the calibration of particle detectors. Moreover, the assessment of sub - 2 nm particles is only important for a small subset of the tested condensation particle counters.

*1°/ The authors argue that a 11 cm Vienna DMA at 30 lpm sheath air can select 1 nm particles with an acceptable resolution. I guess that they have used the DMA 1/40 introduced by Georg Reischl (Winklmayr et al. 1991). Indeed the colleagues are dumbs on this detail.*

The used nano DMA was presented in Winkler et al. (2008). The geometric parameters are listed in Table 1. The reference for the DMA will be corrected and the parameters of the DMA will be presented in the SI. The authors want to re-emphasize that due to the good agreement between measurements of the 50% cut-off diameters with the nano DMA and a high-resolution UDMA for particles with diameters of 1.5 and 2.5 nm (s. SI, Fig. S1), the resolution of the nano DMA was considered good enough for these measurements.
Additionally, the uncertainty of the DMA resolution for sub - 2 nm particles was approximated using the deviation found for 1.2 nm particles (FWHM, p. 665, Fig. 8d, Reischl et al., 1997). The maximum deviation is therefore given by 0.5 nm. Consequently, the deviation was added to and subtracted from selected sub - 2 nm data points presented in this study. This correction was performed for the detection efficiencies of the TSI 3777 (NaCl), the TSI 3788 (NaCl), the TSI 3789 (NaCl) and the TSI $3776^T$ ($BCYO_x$). Figures 1 and 2 show the resulting envelope of the detection efficiency data, the re-calculated 50 % cut-off diameters and curve fits for two exemplary measurements.
The analysis revealed that the related shifts of the 50% cut-off diameters are already covered by the uncertainties given in Table 3 of the manuscript. The only exception is the TSI 3777. As a result, the uncertainties of the 50% cut-off diameters of the TSI 3777 for NaCl and $(NH_4)_2SO_4$ seeds need to be increased to $\pm$ 0.3 nm.

The following changes will be made in the manuscript:

3/25: In the next step, the aerosol enters a custom-made Vienna-type DMA (as presented in Winkler et al. (2008) and referred to as nano DMA, s. SI), where particles are selected according to their electrical mobility.

Table 3: Measurement uncertainties for the TSI 3777 and TSI $3777^T$ increased to $\pm$ 0.3 nm for all seed particle materials due to additional broadening of the transfer function of the classifying DMA.

Supplementary Information: Table 1 added.

Supplementary Information: Figure 1 added. Figure Caption: The Figure shows the envelope of the detection efficiency of the TSI 3777 with NaCl seeds. The black circles and the black line depict the curve presented in the manuscript. The black crosses mark the 50% cut-off diameters as well as their deviation due to diffusional broadening of the transfer function of the DMA (based on Reischl et al. (1997).

*2° / The colleagues are using an X ray charger (to produce ions) for the generated particles before the DMA. This point is critical in my opinion. In the sub 2 nm a charger should be useless and even forbidden when the purity (chemistry in fact) is important. Indeed the figure 2 b, 2 c, 3b, 3c and 4d show results with particles smaller than 2 nm of silver and sodium chloride for example. It is clear that these particles are not pure but dirt. Indeed if you add an ion to a particle to charge it in positive or in negative mode the result has nothing to do with the particle nor with the ion. Ag + Nitrate ion dimer (for example) from the charger is not pure silver particle anymore. Same thing with sodium chloride NaCl + lactic acid (for example) from has nothing to do with sodium chloride. See Maisser et al. (2015) JAS 90, 36-50; Steiner et al. (2014) AST 48,3 261-270 for the chemical composition of ions produced in chargers. I would suggest to the authors to be careful and warn the readers concerning the sub 2 nm results with a charger. High resolution DMAs, electrospray source atomization and evaporation condensation of vapors from hot wires are for the moment the cleanest methods for the generation of clean sub 2 nm. Clean means pure chemistry. Indeed the wire generator is much cleaner than an oven because the hottest point is the wire itself. It's not the case inside an oven. The particles from the wire generator are on the other hand self-charged. It's not the case when an oven is used.*

We agree with the reviewer that the charging process might alter the chemical composition of the aerosol, which might be especially important for particles with diameters smaller than 2 nm. The suggested publications will be cited in order to warn the readers about the mentioned effects.

The exact chemical composition of the produced particles is not known. Consequently, our argumentation is not based on general statements concerning the solubility of seed particles in certain working fluids. The effects might be similar to solubility. Taking into account the lack of information on the chemical purity, we linked our observations to the seed particle materials. We thereby refer to the parent substances of known purity. Nevertheless, the Ag - containing chemical compounds, that are produced with the tube furnace, seem to be chemically more similar to less polar working fluids like n-butanol. On the other hand, the NaCl - containing compounds are activated better in very polar working fluids like DEG. This statement is valid without knowing the exact chemical composition of the seeds as long as reproducible conditions during the generation process can be assumed. The presented results can be interpreted indirectly by knowing the shifts in the 50% cut-off diameter. Another aspect to take into account is the following: Contamination of the seed particles is not generally linked to a change in their polarity. Consequently, a chemical compound composed of NaCl and, as mentioned, lactic acid, could still exhibit a higher polarity compared to another compound consisting of Ag and some contaminating compounds.

We therefore conclude that data needs to be interpreted by pointing out potential sources of chemical impurities. The overall dependencies, as stated, are connected to the used seed particle materials and the used particle generation methods. Tube furnaces are commonly used for seed particle generation, especially for the calibration of particle detectors (e.g. Kangasluoma, 2013; Wimmer, 2013). The presented study provides further insights into this specific calibration methods. Subsection 3.6 and Section 4 will be adapted accordingly.

The following changes will be made in the manuscript:

8/2: Furthermore, the interpretation of the results of seed particles with diameters < 2 nm is restricted to the knowledge of the exact composition of the seed particle material and can not be extended to the composition of the produced aerosol particles. Previous research has shown that the use of tube furnaces and aerosol chargers based on radioactivity cause impurities of the produced aerosol particles (Steiner, 2014; Maisser 2015).

9/24: Future research needs to focus on the extension and verification of the presented results for sub-2 nm particles with known composition. Hence, the presented study suggests a novel and improved approach in determining counting efficiencies of CPCs when a calibration with a variety of seed particles is not available.

*I have few other small details about the work. 1° / The following previous works should appear in the introduction of the paper to my opinion Seto et al (1997) https://doi.org/10.1063/1.474510 Gamero & Fernandez de la Mora https://doi.org/10.1016/S0021-8502(99)00555-8 Attoui 2018 : https://doi.org/10.1016/j.jaerosci.2018.08.005*

The mentioned publications will be included as references in the "Introduction" - section.

The following changes will be made in the manuscript:

2/5: Since the introduction of Aitken's "Dust Counter" towards the end of the 19th century (Aitken, 1888), particle counters capable of detecting smaller and smaller aerosol particles have been developed (Stolzenburg and McMurry, 1991; Seto et al., 1997; Gamero-Castano and Fernandez de la Mora, 1999; McMurry, 2000; Sgro and Fernandez de la Mora, 2004; Vanhanen et al., 2011; Kangasluoma and Attoui, 2019).

3/4: Additionally, previous studies have shown that the detection efficiency of CPCs for sub-2 nm particles can be improved by tuning the instruments settings (Barmpounis et al., 2017; Attoui, 2018).

*2° / I don't understand very well why the authors are using two identical set ups. What is the benefit of the DMA working at 19.5 lpm? Why not 20 lpm by the way?*

The presented schematics of the setups are similar but not exactly the same. Due to the use of the TSI 3777 the flow rates had to be increased and the sheath air flow rates of the DMA were adjusted. The flow rates were maintained by a critical orifice and frequently checked with a TSI 4000 Series Flow Meter. Accordingly the measured sheath air flow rate was 19.5 ± 0.4 lpm. When using the TSI 3777 the sheath flow of the nano DMA was increased to 30 ± 0.6 lpm. The sheath air flow rates were results of the used critical orifices.

*3° / It will be good to give the geometrical parameters of the DMAs.*

The geometrical parameters of the used DMA will be stated in a table (s. answer to major comment 1°/).

*4° / The authors are giving the resolution of their DMA but nothing about the flowrate nor the size of the particles they have used for the measurement of the resolution. Was it done with a tandem DMA by the way?*

The resolution of the DMA was not measured but approximated using Equation 2 and the flow rates are mentioned in the manuscript. Further consideration about the DMA's resolution was based on the results of Reischl et al. (1997), as stated above.

*5° / The equation 2 is useless since the authors are talking about particles down to 1 nm (as small as !) where the diffusion is very active adversely to what Rick Flagan was telling in the cited paper.*

The validity of the cited equation will be addressed.

The following changes will be made in the manuscript:

3/30: The limiting resolution throughout our experiments is therefore approximated by 0.15. It is important to note that in the 1 - 2 nm size range diffusional broadening can degrade the DMA resolution (Jiang et al., 2011). The selected aerosol might be polydisperse and the selected particle diameter must be seen as an average diameter with an envelope (s. Fig. S1, SI). In our study this envelope is given by ± 0.5 nm, based on a maximum uncertainty approximation by taking into account the results of Reischl et al. (1997). The diffusional broadening of the transfer function of the nano DMA between 1 and 5 nm was also presented by Winkler et al. (2008). In their study the authors calculated a geometric standard deviation of approximately 1.07 for a size-classified output aerosol composed of particles with a mean diameter of 1 nm.

*6° / In the figure 2 the diameter is given in standard number format. It's not the case in the figures 3 and 4. Please use the same format. I would suggest standard format rather than scientific format.*

The mentioned axes of Figure 3 and Figure 4 will be adapted accordingly in the final version of the manuscript.

The following changes will be made in the manuscript:

Figure 3: Abscissa scaled linearly.

Figure 4: Abscissa scaled linearly.

*7° / Susan Hering and colleagues as published in a paper the experimental results of the versatile CPC commercialized by TSI. That paper is the most adequate reference to the instrument. The paper is cited on the TSI website. Susanne V. Hering, Gregory S. Lewis, Steven R. Spielman, Arantzazu Eiguren-Fernandez, Nathan M. Kreisberg, Chongai Kuang & Michel Attoui (2017) Detection near 1-nm with a laminar-flow, waterbased condensation particle counter, Aerosol Science and Technology, 51:3, 354-362.*

The mentioned publication will be cited as a reference for the TSI 3789.

The following changes will be made in the manuscript:

4/35: Lastly, the TSI 3789 (V-WCPC 3789) is based on a three-step principle: a cool conditioner, a warm initiator and a cooler moderator (Hering, 2017).

| | |
|---|---|
| Inner Diameter $R_i$ | 0.0175 m |
| Outer Diameter $R_o$ | 0.0241 m |
| Length $L$ | 0.0150 m |

Table 1: The table summarizes the geometrical parameters of the used nano DMA.

[Figure]

Figure 1: The Figure shows the envelope of the detection efficiency of the TSI 3777 with NaCl seeds.

[Figure]

Figure 2: The Figure shows the envelope of the detection efficiency of the TSI 3789 with NaCl seeds.

**Final Author's Response to Referee #3**

By P. J. Wlasits

peter.wlasits@univie.ac.at

We thank Referee #3 for the critical assessment of our work and the helpful comments. In the following we address the comments point by point:

*General: Please use a different method to denote the tuned instrument rather than an asterisk, as this normally implies a footnote. Suggest superscript-'T'*

The suggestion will be implemented as stated.

The following changes will be made:

5/4, 6/19, 7/6, 7/8, 8/5, 8/7, 8/18, 8/26, Fig. 4, Table 3, Table 4, Fig. 7 and Fig. S1: Asterisks replaced by T's.

*Page 3, line 12: The purities and grades of all of the chemical stocks should be stated, including the solvent used for the BCY solution. Also state the concentration of the BCY solution.*

The BCY was used as purchased and was not diluted. The purity of the used BCY will be added and the word 'BCY solution' will be replaced by 'BCY' in order to avoid confusion.

The following changes will be made:

3/11: The counting efficiencies of the CPCs were measured using four different types of seed particles, generated from sodium chloride (NaCl, pro analysi, Merck KGaA, Darmstadt, Germany), silver (Ag, wool for elemental analysis, CAS No.: 7440-22-4, Merck KGaA, Darmstadt, Germany), ammonium sulfate ($(NH_4)_2SO_4$, pro analysi, Merck KGaA, Darmstadt, Germany) and $\beta$-caryophyllene ($C_{15}H_{24}$, BCY, $\geq 80\%$, CAS No.: 87-44-5, Sigma Aldrich, St. Louis, USA).

4/18: The seeds were produced by evaporating BCY into a clean airstream and subsequently mixing it with ozone, allowing the ozonolysis reaction to take place inside the flow tube.

Fig.1 / Caption Fig.1: $C_{15}H_{24}O_x$ seeds were generated using BCY (BC), an ozone generator (OG) and a flow tube.

*Page 4, line 18: State the method used to generate ozone and control the concentration*

An UV lamp was used to generate ozone. The concentrations were not monitored due to previous experiments on the performance of the lamp.

The following sentence will be added:

4/19: Ozone was generated using a custom made UV-lamp with adjustable intensity. An intensity/ozone calibration was performed prior to the experiments with an ozone monitor (ThemoFischer Scientific Model i49), suggesting that the ozone concentrations were in the range of 100-500 ppb.

*Page 5, line 24: Please do not use the word 'saturates', as this could cause confusion.*

In order to avoid confusion, the word 'saturates' will be replaced by the word 'reaches'.

The following changes will be made:

5/29: On the contrary, the detection efficiency of the TSI 3789 reaches 1 at about 20 nm.

*Page 6, line 5: Is this not related to the particle's solubility rather than polarity?*

We suspect that especially in the sub-10 nm size range charge effects might play a crucial role during the activation of seed particles (s. Figure 7 and 8/32). In order to properly include that into our argumentation, we refrained from using the word 'solubility'. Nevertheless, the observed effects are very similar to dissolution processes.

*Page 6, line 19: The word 'astoundingly' isn't particularly scientific. Please describe what aspect was unusual or unexpected.*

Since the observation is discussed in the following lines, the word 'astoundingly' will be deleted.

The following changes will be made:

6/19: The TSI $3776^T$ shows barely no composition dependence of the 50% cut-off diameters, thereby confirming the results of Brilke et al. (2020) for Ag seeds.

*Page 7, line 11: The phrase "The effect of just readjusting temperatures can be clearly seen too by. . ." is very clumsy. Please reword.*

The following changes will be made:

7/11: Fig. 6a shows the effect of changing the temperature settings and the inlet flow of the TSI 3776. The effect of just changing the temperature settings is shown in Fig. 6b and Fig. 6c (TSI 3777 and TSI 3789).

*Page 8, line 32: Remove brackets around the reported statistic*

The brackets will be removed throughout the text.

The following changes will be made:

5/17, 5/18, 5/19, 5/24, 6/4 and 8/32: Brackets around the statistics removed.

*Page 10, line 6: The statement about the work being performed independently should really come under the competing interests statement. Could the same statement about TSI be extended to Airmodus?*

The presented work has been performed without funding from any company. The statement is true for TSI Inc as well as Airmodus Ltd. The related statement will be moved into the "Competing Interest" - Section and Airmodus Inc. will be included.

The following corrections will be made:

[revised manuscript text omitted]

[1] Iida et al. (2009), Aerosol Sci. Tech.,43:1

[2] National Center for Biotechnology Information. PubChem Database. Water, CID=962, accessed on 18.11.2019

[3] C. Reichardt and T. Welton (2010), *Solvents and Solvent Effects in Organic Chemistry*, Wiley VCH

**Table S2.** Temperatures of the Tube Furnace: The table displays the minimum and maximum temperatures of the tube furnace that have been set in order to generate particles with mobility diameters between 1 and 25 nm.

| Seed Material | $T_{min}$ [K] | $T_{max}$ [K] |
|---|---|---|
| NaCl | 793.15 | 873.15 |
| Ag | 1073.15 | 1193.15 |
| $(NH_4)_2SO_4$ | 463.15 | 513.15 |

**Table S3.** The table summarizes the geometrical parameters of the used nano DMA.

| | |
|---|---|
| Inner Diameter $R_i$ | 0.0175 m |
| Outer Diameter $R_o$ | 0.0241 m |
| Length $L$ | 0.0150 m |

[Figure]

**Figure S1.** The Figure shows the envelope of the detection efficiency of the TSI 3777 with NaCl seeds. The black circles and the black line depict the curve presented in the manuscript. The black crosses mark the 50% cut-off diameters as well as their deviation due to diffusional broadening of the transfer function of the DMA (based on Reischl et al. (1997).

[Figure]

**Figure S2.** Comparison between nano DMA and UDMA Measurements: The Figure shows the 50 % cut-off diameters for negatively charged Ag and NaCl seeds measured with the TSI $3776^T$ and the TSI 3777. The spherical markers refer to data obtained by using the a nano DMA as described in Section 2. The triangles represent data measured with an UDMA setup (s. Brilke et al., 2020).

[Figure]

**Figure S3.** Detection Efficiencies and Seed Particle Material: The Figure shows the detection efficiencies based on different working fluids as a function of the electrical mobility equivalent diameter. Different colors correspond to different particle counters and every plot is related to a different seed particle material: BCY (Panel a), ammonium sulfate (Panel b), silver (Panel c) and sodium chloride (Panel d).

[Figure]

**Figure S4.** Detection Efficiencies of the TSI 3789: The Figure shows the detection efficiencies for different seed particle materials as a function of the electrical mobility equivalent diameter. Different colors correspond to different seed particles.

[Figure]

**Figure S5.** Corrected Detection Efficiencies: The Figure shows detection efficiency data of the TSI 3789 for $BCYO_x$ seeds (Panel a), $(NH_4)_2SO_4$ seeds (Panel b) and $NaCl$ seeds (Panel c), that have been corrected for diffusional losses.

[Figure]

**Figure S6.** Supersaturation Profiles of Three CPCs: The Figure shows the supersaturation profiles of the TSI 3776 (Panel a), the tuned TSI 3776 (Panel b) and the TSI 3772 (Panel c). The abscissa corresponds to the axial distance on the centerline of the condenser and the ordinate depicts the radial distance. The colors correspond to different saturation ratios.

[Figure]

**Figure S7.** Saturation Ratio in the Condensers: The Figure shows the centerline saturation ratio as a function of the axial distance for the TSI 3776 (Panel a), the tuned TSI 3776 (Panel a) and the TSI 3772 (Panel b).